# Effects of Polyethylene Microplastics in Agricultural Soil on *Eisenia fetida* (Annelida: Oligochaeta) Behavior, Biomass, and Mortality

Milica Baloš [1], Aleksandra Petrović [1,*], Aleksandra Tubić [2], Tijana Zeremski [3], Sonja Gvozdenac [3], Dejan Supić [4] and Vojislava Bursić [1]

1   Faculty of Agriculture, University of Novi Sad, Trg Dositeja Obradovića 8, 21000 Novi Sad, Serbia; cucuzm@yahoo.com (M.B.); bursicv@polj.uns.ac.rs (V.B.)
2   Faculty of Sciences, University of Novi Sad, Trg Dositeja Obradovića 3, 21000 Novi Sad, Serbia; aleksandra.tubic@dh.uns.ac.rs
3   Institute of Field and Vegetable Crops, Maksima Gorkog 30, 21000 Novi Sad, Serbia; tijana.zeremski@ifvcns.ns.ac.rs (T.Z.); sonja.gvozdenac@ifvcns.ns.ac.rs (S.G.)
4   Faculty of Ecological Agriculture, University Educons, Vojvode Putnika 87, 21208 Sremska Kamenica, Serbia; dejan.supic@educons.edu.rs
*   Correspondence: aleksandra.petrovic@polj.edu.rs

**Abstract:** The presence of microplastic particles in agroecosystems has profound implications for soil quality, crop yield, and soil biota. Earthworms are widely recognized as valuable soil bioindicators due to their abundance, fast reproduction, and easy manipulation. The aim of this study was to observe *Eisenia fetida* avoidance behavior and changes in biomass and mortality rate in soil samples spiked with polyethylene microplastic particles. Three types of soil sampled from the agricultural fields ("Banat 1", "Banat 2", and "Bačka") were tested, as well as three microplastic concentrations (0.1, 0.2, and 0.3%). The calculated avoidance percentages ranged from 18.67% for "Banat 1" and 23.70% for "Banat 2" to 27.40% in the case of "Bačka" soil samples. Generally, *E. fetida* specimens avoided the sections with plastic in all bioassays: 38.42% of the earthworms were in the chamber section that contained microplastics, as opposed to 61.58% in the control section. The changes in the earthworms' post-test biomasses were directly proportional to the number of surviving earthworms, with the highest loss in "Bačka" soil samples with 0.3% MPs (−53.05%). The highest mortality rate (46%) was noted in "Bačka" soil samples spiked with the highest concentration of microplastic particles.

**Keywords:** microplastic; soil; *Eisenia fetida*; avoidance test; biomass; mortality; bioassays; earthworms

## 1. Introduction

In recent decades, plastic waste has been reported to be one of the most important factors that have significantly contributed to intensive and complex environmental pollution [1]. In this regard, the term "plastic" includes a wide range of mixtures, primarily including polyethylene (PE), polypropylene (PP), polyethylene terephthalate (PET), and polyvinyl chloride (PVC) [2]. Plastic materials are subject to the heavy influence of various environmental factors in aquatic and terrestrial ecosystems, which leads to their decomposition into smaller particles specified as micro- (MP) and nanoplastic (NP). The definition of NP and MP follows the nomenclature of the International System of Units, and they are typically considered to be 1–100 nm and 1–5000 μm in size, respectively [3]. These particles are ubiquitous in the environment, and they are easily transported by various mechanisms [4]. Recently, terrestrial ecosystems, especially agroecosystems, have gained increasing recognition for generating and absorbing plastic pollution [5,6]. These discoveries have highlighted the necessity of investigating the effects of MPs on the agroecosystems' health and sustainability. It is important to note that agricultural fields constitute nearly

half of the Earth's terrain, and the soil quality is crucial for ensuring global food safety and crop production [7].

The application of plastic materials in agricultural production has a long history. In 1948, E. M. Emmert, a horticulture scientist and professor at the University of Kentucky, noticed a cost-effective alternative to glass for his greenhouse flowers. He initially experimented with cellulose acetate film, and later replaced it with PE mulch film. This marked the first application of plastic material in agriculture, and its usage has been steadily increasing since then [8]. The most common sources of MPs in agroecosystems are agricultural films, pesticides and mineral fertilizers packages, compost, sewage sludge, tire abrasion from machinery, and atmospheric deposition [9–12].

Globally, plastic mulching has been used on approximately 20 million hectares of agricultural land. The most commonly used types of plastic in this context are PVC, PP, PE, and ethylene–vinyl acetate copolymer (EVA copolymer). According to Huang et al. [13], approximately 4.5 million tons of plastics were used for mulching in 2019, and this figure is expected to reach 5.6 million tons by 2030. Recently, mulching has become a widespread practice due to its ability to improve fruit quality, increase yields, and enhance water utilization. However, prolonged coverage and inadequate recovery lead to a significant accumulation of plastic residues in the soil. Over time, these residues degrade into smaller plastic particles, which can reduce seed quality, disrupt soil structure, and have negative effects on soil organisms [14]. Furthermore, the use of mineral fertilizers and plant protection products is continuously increasing. Plastic waste originating from mineral fertilizer sacks, pesticide packaging, and mulch films represent significant sources of MPs in agricultural soil, since their recycling process is still challenging. Although compost has been considered an environmentally friendly soil amendment, if it contains MPs, it becomes a long-term contaminant in the environment [15]. Finally, an important MPs source in agriculture is sewage sludge. It is rich in organic matter and essential nutrients and is frequently used as a fertilizer to improve soil productivity. Sewage sludge is derived from domestic wastewater, which usually contains microbeads from cosmetic products, polymer fibers from washing clothes, industrial waste, and rubber wear, and therefore contains significant amounts of MPs [15].

Although the United Nations Environment Programme (UNEP) listed MP soil pollution among the top ten environmental issues in 2014, this problem still has considerably less scientific and public attention than MP pollutions detected in aquatic ecosystems. However, the amount of MPs released into the soil is 4 to 23 times greater than those found in the aquatic ecosystems [16]. MPs affect the soil's physical, chemical, and biological properties, including soil density, microbial activities, and plant growth and maturity [17]. Furthermore, due to their small size, hydrophobic nature, and large specific surface area, MPs can adsorb toxic substances from the soil, including heavy metals [18]. Consequently, MPs and heavy metals may cause a synergistic environmental pollution, resulting in potentially harmful impacts on terrestrial organisms. MPs in soil can be readily consumed and digested by animals [19], which directly affects their physiological condition, nutrition, metabolism, and mortality. Indirectly, it can impact reproduction and population attributes, such as population density (biomass) and growth, spatial distribution, age, and their presence in specific habitats [20].

Earthworms are often proposed as bioindicators for assessing soil quality due to their integral role in terrestrial ecosystems. They are widely recognized as valuable indicators used to assess critical thresholds for determining soil pollutants [21] due to their abundance and fast reproduction [22]. The earthworms' presence and activities are commonly associated with favorable soil conditions, indicating good soil quality. Through various activities in the soil, these species decompose organic matter, convert nutrients into forms easily accessible to plants, microorganisms, and other animal species, contribute to the structural development of soil aggregates, aeration, and porosity, and improve water drainage, as well as root penetration [23].

In the past decade, earthworms were frequently used as the accurate bioindicators of soil pollution by various MP and NP types. The most common species used in these

studies were *Lumbricus terrestris* Linnaeus, 1758 and *Eisenia fetida* (Savigny, 1826) [24]. The published studies were aimed at investigating issues such as the direct influence of MPs on earthworms' condition, reproduction, growth, and behavior through ingestion and consequently accumulation [25–27]; the impact of MPs of different types, sizes, and concentrations [28,29]; MPs' exposure and degradation routes [30]; indirect effects regarding the combined interaction of MPs and soil [31,32], as well as possible long-term ecological significances and consequences [33,34]. Since there are more than 5000 different types of plastic materials used for different purposes [35], we wanted to test those that have been frequently used in agricultural production (e.g., pesticide packaging). Therefore, the aim of this study was to monitor *Eisenia fetida* avoidance behavior and changes in biomass and mortality rate when introduced to soil samples spiked with MP particles.

## 2. Materials and Methods

### 2.1. Soil and Microplastics

The soil samples were collected from three geographically distant agricultural fields of the Institute of Field and Vegetable Crops, Novi Sad (Vojvodina, Northern Province of the Republic of Serbia), using a shovel, from the 10–20 cm depth. The soil samples were labeled as "Banat 1", "Banat 2", and "Bačka". A year prior to sampling, the fields were not treated with any agrochemicals. Therefore, the soil was not sterilized, and the natural condition of the soil was maintained to avoid other stressors in the bioassays.

The collected soil samples were air-dried and milled to <2 mm particle size, according to ISO 11464: 2004 [36]. The pH values of the soil suspension in water and in 1M KCl, organic matter content, as well as free $CaCO_3$ content were analyzed using standardized ISO methods: ISO 10390: 1994 [37], ISO 14235: 1998 [38], and ISO 10693:1995 [39], respectively. Particle size distribution was estimated after particle fractionation in the following size fractions: coarse sand (200–2000 μm), fine sand (20–200 μm), silt (20 μm), and clay (2 μm). The fractionation was performed by the sieving and pipetting method according to Van Reeuwijk [40]. Total nitrogen was determined by CHNS elemental analysis using a VarioEL III analyzer (Elementar, Germany), according to the AOAC Official Method 972.43: 2000 [41]. Readily available phosphorus P (AL) in soil was determined by extraction of ammonium lactate [42], whereby detection was performed at the wavelength of 830 nm in a UV/VIS spectrophotometer, Cary 60 (Agilent Technologies, Waldbronn, Germany) using the phosphomolybdate blue method [43]. All used chemicals were purchased from J.T. Baker (Mallinckrodt Baker, Inc., Phillipsburg, NJ, USA).

The MPs used in all bioassays originated from the agrochemical packaging collected from the agricultural fields in the vicinity of the soil sampling sites, and they were finely cut for the purposes of the experiment. MP particles were less than 5 mm in diameter, irregularly shaped, and heterogeneous. Fourier-transform infrared spectroscopy (FTIR) (Nicolet iS20 FTIR spectrometer, Thermo Fisher Scientific, Waltham, MA, USA) was used to detect the MP type.

### 2.2. Earthworms

The colonies of *Eisenia fetida* (Savigny, 1826) were purchased from the local compost and earthworm farm (Apatin, Serbia). Only adult earthworms with fully developed clitellum and individual mass above 300 mg were used in both types of bioassays. Prior to introduction into the test containers, the earthworms were measured using the analytical scale Kern 440-47N (KERN& SOHN GmbH, Lörrach, Germany), rinsed with deionized water, and gently dried with filter paper.

### 2.3. Avoidance Bioassays

The influence of MP particles on the earthworms' behavior was observed in avoidance tests [44]. For the purpose of these tests, the soil samples (500 g) were spiked with MP particles (0.5, 1, and 1.5 g) in glass beakers and mixed evenly, and the total amount was poured into the specific container compartments. Thus, the MP concentrations were 0.1, 0.2,

and 0.3% (*w/w*) with 5 replicates for each treatment and each soil type [45]. Soil without MPs served as the control group (500 g). In this way, prepared soil samples were aged for 7 days before starting the experiments [46].

For the purpose of the experiment, the two-section chamber containers made of stainless steel with capacities of 1 L were used. The containers allowed sufficient light access and gaseous exchange between the soil and the air, as they were covered with perforated transparent stretch foil, fixed to the sides with the tape in order to prevent the earthworms from escaping.

At the beginning of the bioassay, the containers were split into two equal sections using the vertically introduced plastic divider. The left compartment of the container was filled with MP-spiked soil (test soil, 500 g), and the right side with the control soil (no MP added, 500 g). Both soil heights were leveled equally, at approximately 50 to 60 mm. Thereafter, the plastic divider was removed, and 10 earthworms were placed on the separating line. Their burrowing activity was quick in each case, as they used the slit left by the plastic separator as a starting point. All containers were kept in the same environmental chamber (20 ± 2 °C, 40–42% relative air humidity) (measured with Profi-Thermo-/Hygrometer HygroLogg Pro, TFA Dostmann GmbH & Co. KG, Wertheim, Germany) under light–dark cycles of 16–8 h. The bioassay lasted for 48 h. During that time, additional food sources for the earthworms were not supplemented and the soil was not watered.

At the end of the bioassay, the test and control soil compartments were separated again using the same plastic divider. The dividers were inserted in the environmental chambers in order to avoid possible *E. fetida* movement due to the temperature and humidity changes. The number of available earthworms was calculated separately for both compartments in the container. The specimens cut by divider insertion were calculated as either 0.5 if the length of the body was equal on both sides, or as 1 if the remaining part of the body was longer or if it was determined as a prostomium bearer. Missing earthworms were considered to have died and disintegrated during the test period.

The avoidance expressed as a percentage (A) was calculated according to the equation

$$A\ (\%) = ((n_c - n_t)/N) \times 100 \tag{1}$$

where A is avoidance (%); $n_c$ is the number of earthworms in the control soil, $n_t$ is the number of earthworms in the test soil, and N is the total number of earthworms in the container. The bioassay was considered inaccurate and invalid if the number of dead or missing earthworms was more than 1 individual per replicate.

### 2.4. Biomass and Mortality Bioassays

The previously described protocol was used in order to prepare soil samples for biomass and mortality bioassays as well, although the total amount of soil was 1000 g per replicate. Same as in previous tests, the MP concentrations were 0.1, 0.2, and 0.3% (*w/w*) with 5 replicates for each treatment and soil type [45]. Soil without MPs served as the control group (1000 g). The soil samples were placed in small plastic flower pots perforated at the bottom in order to provide sufficient water drainage. For each replicate, 10 earthworms were collected from the compost, gently rinsed with deionized water, and dried with filter paper. Each batch of 10 earthworms was measured using the analytical scale Kern 440-47N (KERN& SOHN GmbH, Germany) in order to obtain the initial biomass values. The earthworms were gently placed in a hand-made shallow hole in the pot filled with previously prepared soil samples and then covered with a small amount of soil.

All containers were kept in the same environmental chamber (20 ± 2 °C, 40–42% relative air humidity) (measured with Profi-Thermo-/Hygrometer HygroLogg Pro, TFA Dostmann GmbH & Co. KG), under light–dark cycles of 16–8 h. The bioassays lasted for 10 days, and during that time, the earthworms were not additionally fed, but the soil was regularly watered with 100 mL of tap water every other day.

On the 11th day, the contents of the pots were poured on a sheet of white paper, the soil was thoroughly prospected, and all earthworms were gently separated, rinsed

again with deionized water, dried with filter paper, and measured on the analytical scale in order to obtain final biomass values. The changes in biomass were compared using individually calculated mass (total biomass divided by the number of surviving specimens). Mortality was calculated as a percentage of dead or missing earthworms in relation to the initial population (of 10). The earthworms were considered dead if there was no movement at all, even after touching or gently moving, or when they were dried up. In order to compare biomass and mortality values, Schneider-Orelli's formula [47] for corrected mortality was applied:

$$M_{corrected}\ (\%) = (\%M_t - \%M_c)/(100 - \%M_c) \times 100 \qquad (2)$$

where $M_{corrected}$ is the corrected earthworm mortality percentage, $\%M_t$ is the earthworm mortality percentage in soil samples spiked with MPs, $\%M_c$ is the earthworm mortality percentage in soil without MPs.

### 2.5. Statistical Analyses

The percentages were mathematically calculated using Microsoft Office Excel 2019 (Microsoft Office Standard, 2019, University License). The same software was used to draw the chart presented in Section 3.3. The obtained results for avoidance bioassays were statistically analyzed using Wilcoxon matched pairs tests. The analysis of variance (ANOVA) and Fisher's Least Significant Difference (LSD) test were used to analyze avoidance, biomass changes, and mortality percentages. Statistical significance was observed for $p < 0.05$ (interpreted as a high significance) and $p < 0.01$ (interpreted as a very high significance). All statistical analyses were performed using Statistica 14.0.0.15 (TIBCO Software Inc., Palo Alto, CA, USA, University License).

## 3. Results and Discussion

### 3.1. Soil Properties and Microplastic Characteristics

The physicochemical properties of the tested soil samples are presented in Tables 1 and 2. Soil samples labeled as "Bačka" had good chemical properties and light texture due to a high percentage of coarse and fine sand. "Banat 1" had a similar texture to "Banat 2", but the latter had a slightly heavier mechanical composition. The obtained results indicated that the physicochemical properties of "Banat 1" were near the optimal value ranges for earthworms, which was in accordance with the results published by Gebremeskel Weldmichael et al. [48] and Weldmichael et al. [48].

**Table 1.** Physical characteristics of tested soil samples.

| Sample | Coarse Sand (%) 2–0.2 mm | Fine Sand (%) 0.2–0.02 mm | Silt (%) 0.02–0.002 mm | Clay (%) <0.002 mm | Soil Texture Class * |
|---|---|---|---|---|---|
| Banat 1 | 1.26 | 35.50 | 30.20 | 33.04 | Loamy clay |
| Banat 2 | 0.43 | 22.45 | 32.92 | 44.20 | Loamy clay |
| Bačka | 36.43 | 56.25 | 4.84 | 2.48 | Loamy fine sand |

* According to the International Union of Soil Sciences.

**Table 2.** Chemical characteristics of tested soil samples.

| Sample | pH in KCl | pH in $H_2O$ | $CaCO_3$ (%) | Organic Matter (%) | Total N (%) | AL-$P_2O_5$ mg/100 g | AL-$K_2O$ mg/100 g |
|---|---|---|---|---|---|---|---|
| Banat 1 | 7.32 | 8.33 | 10.66 | 3.11 | 0.213 | 18.66 | 42.69 |
| Banat 2 | 6.85 | 7.83 | 1.11 | 3.27 | 0.224 | 57.08 | 72.02 |
| Bačka | 6.86 | 7.27 | 2.27 | 5.17 | 0.332 | 18.25 | 31.74 |

FTIR characterized all MP particles as polyethylene (PE) (Figure 1).

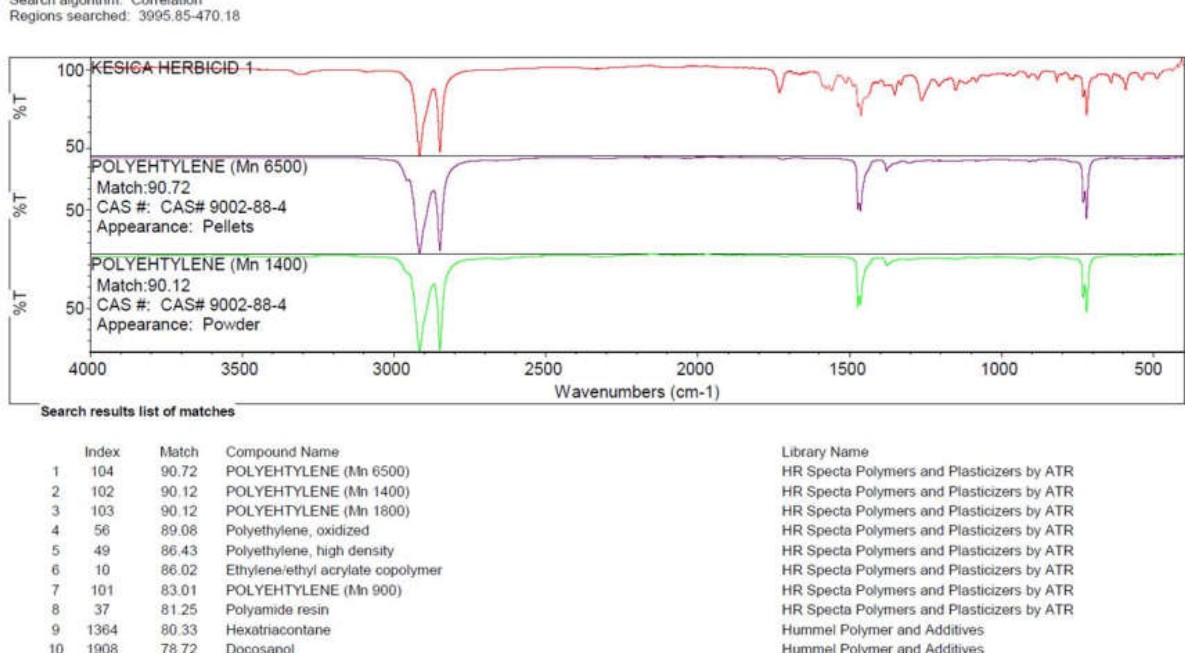

**Figure 1.** The results of FTIR performed for detection of MP particles' type.

Polyethylene, commonly known as polythene (PE) or by its IUPAC (International Union of Pure and Applied Chemistry) name—polyethene—or poly(methylene), is one of the most used plastic materials. In 2017, the global production of PE resins exceeded 100 million tons, constituting 34% of the total plastics market [49]. The key features that make PE attractive include its affordability, excellent electrical insulation across a broad frequency range, impressive chemical resistance, good processability, toughness, flexibility, and in certain grades, transparency in thin films [50]. PE is not biodegradable, and due to its density of 0.85–0.98 g/cm$^3$, it floats on the surface of water. It has been frequently used in packaging, the production of bags, and in wire insulation and bottles. The degradation of this high-molecular-weight polymer in the environment is very slow and represents the result of the combined effects of photo- and thermo-oxidative degradation coupled with biological activities. According to Hakkarainen and Albertsson [51], the initial and rate-determining step in biodegradation is abiotic oxidation.

PE microplastic particles affect the growth of crop plants, as evidenced by their impact on soil properties and the soil biota in agroecosystems [52]. PE microplastics have diverse effects on soil-dwelling organisms. Earthworms facilitate the transport of PE in soil actively or passively through different activities: burrowing, feeding processes, defecation, and adherence to cutaneous mucus [53]. Consequently, these processes usually induce histopathological injuries and immune system responses in earthworms [54].

### 3.2. E. fetida Avoidance Bioassays

The results of the two-section chamber avoidance test are presented in Table 3.

Generally, *E. fetida* specimens avoided the section with MP in all bioassays, with an average of 3.8 earthworms in the section of the chamber that contained MP, in contrast to the 6.09 earthworms in the section that served as a control. According to Windsor et al. [55], MP particles could affect earthworms similarly to aquatic worm species, where the most frequent consequences involve the blockage and abrasion of the digestive tract, resulting in limited nutrient bioavailability and absorption, reduced growth, and ultimately jeopardized organism survival [56].

**Table 3.** The results of the two-section chamber avoidance test.

| Sample | MP Conc. (%) | Average Number of *E. fetida* | | Mortality (%) | A (%) | Average Avoidance (%) |
| | | Left Section (MP) | Right Section (Control) | | | |
| --- | --- | --- | --- | --- | --- | --- |
| Banat 1 | 0.10 | 4.70 | 5.30 | 0.00 | 6.00 | |
| | 0.20 | 4.10 | 5.90 | 0.00 | 18.00 | 18.67 |
| | 0.30 | 3.40 | 6.60 | 0.00 | 32.00 | |
| Banat 2 | 0.10 | 4.50 | 5.50 | 0.00 | 10.00 | |
| | 0.20 | 3.60 | 6.40 | 0.00 | 28.00 | 23.70 |
| | 0.30 | 3.30 | 6.50 | 2.00 | 33.11 | |
| Bačka | 0.10 | 4.30 | 5.50 | 2.00 | 12.20 | |
| | 0.20 | 3.20 | 6.60 | 2.00 | 34.67 | 27.40 |
| | 0.30 | 3.10 | 6.50 | 4.00 | 35.33 | |

The obtained results were statistically analyzed by a Wilcoxon matched pairs test, presented in Table 4. Very high statistical significances were calculated for all bioassays, with the highest values observed for "Bačka" soil samples.

**Table 4.** The results of Wilcoxon matched pairs tests.

| Sample | Valid N | T | Z | *p*-Value |
| --- | --- | --- | --- | --- |
| Banat 1 | 12 | 0.00 | 3.059412 | 0.002218 * |
| Banat 2 | 13 | 0.00 | 3.179797 | 0.001474 * |
| Bačka | 14 | 0.00 | 3.295765 | 0.000982 * |

* Marked tests (*) are significant at $p < 0.01$.

Nevertheless, ANOVA did not show any statistical significances regarding avoidance percentage (A) as a dependent variable and soil type as a categorical predictor ($p_s = 0.315338$, for $p < 0.05$), but it emphasized very high statistical differences regarding MP concentrations as the categorical factor ($p_c = 0.0000007$, for $p < 0.01$). Fisher's LSD test marked statistically significant differences between the lowest MP concentration (0.1%) and the other two (0.2 and 0.3%).

The highest mortality percentage was detected in Bačka soil samples with the highest MP concentration (0.3%). These results remained at the previously established range, where all replicates had less than one dead or missing earthworm. "Bačka" soil samples matched the sandy soil texture class that contains more macropores, such that consequently, MP particles tended to shift more readily. According to Medyńska-Juraszek and Szczepańska [57], this could enhance a swift downward movement of MP particles into deeper soil layers. The same bioassay ("Bačka" with 0.3% MP) had the highest avoidance percentage, and the lowest was observed for soil samples "Banat 1" with 0.1% MP. On average, the avoidance percentages ranged from 18.67% for "Banat 1" to 27.40% in the case of "Bačka" soil samples (Table 3). The obtained results are supported by the fact that earthworms are hydrophilic, and, therefore, dry sandy soils are not suitable habitats for them. Sandy soils tend to dry out more rapidly and contain limited nutrients and organic matter; therefore, the survival of earthworms in these soil types is very difficult [48]. Additionally, these soil types are not suitable for earthworms due to the sand grains' abrasiveness, which induces cuticle damage [48].

*3.3. E. fetida Biomass and Mortality Bioassays*

The average mass of individual earthworms in pre-test measured biomasses ranged from 399.88 mg (Banat 2, control group) to 442.72 mg (Bačka, 0.1% MP). At the end of the experiment, the highest mortality percentage was observed in soil samples collected from the Bačka locality spiked with the highest concentration of PE MP particles (0.3%) (Table 5).

**Table 5.** The results of the biomass (with standard deviation—SD) and mortality (M) bioassays.

| Sample | MP Conc. (%) | Pre-Test Number | Pre-Test Biomass (g) | SD | Post-Test Number | Post-Test Biomass (g) | SD | Biomass (%) | SD | Biomass Change (of 100%) | M (%) | M$_{corr.}$ (%) |
|---|---|---|---|---|---|---|---|---|---|---|---|---|
| Banat 1 | - | 10 | 4.274 | 0.284 | 9.8 | 4.187 | 0.263 | 98.20 | 6.747 | −1.80 | 2.00 | |
| | 0.10 | 10 | 4.099 | 0.203 | 9.8 | 3.989 | 0.255 | 97.31 | 3.778 | −2.69 | 2.00 | 0.00 |
| | 0.20 | 10 | 4.099 | 0.166 | 9.8 | 3.992 | 0.355 | 97.29 | 6.007 | −2.71 | 2.00 | 0.00 |
| | 0.30 | 10 | 4.100 | 0.103 | 9.4 | 3.772 | 0.241 | 91.97 | 4.668 | −8.03 | 6.00 | 4.08 |
| Banat 2 | - | 10 | 3.999 | 0.293 | 9.8 | 3.923 | 0.425 | 98.16 | 8.395 | −1.84 | 2.00 | |
| | 0.10 | 10 | 4.265 | 0.213 | 9.4 | 3.930 | 0.326 | 92.27 | 8.147 | −7.73 | 6.00 | 4.08 |
| | 0.20 | 10 | 4.070 | 0.188 | 9.2 | 3.638 | 0.438 | 89.41 | 10.388 | −10.59 | 8.00 | 6.12 |
| | 0.30 | 10 | 4.189 | 0.324 | 7.8 | 3.054 | 0.458 | 73.34 | 12.503 | −26.66 | 22.00 | 20.41 |
| Bačka | - | 10 | 4.243 | 0.278 | 7.8 | 3.142 | 0.394 | 74.07 | 8.274 | −25.93 | 22.00 | |
| | 0.10 | 10 | 4.427 | 0.382 | 6.4 | 2.673 | 0.257 | 60.56 | 6.078 | −39.44 | 36.00 | 17.95 |
| | 0.20 | 10 | 4.320 | 0.293 | 6.8 | 2.692 | 0.482 | 58.68 | 8.918 | −41.32 | 32.00 | 12.82 |
| | 0.30 | 10 | 4.290 | 0.459 | 5.4 | 1.805 | 0.617 | 46.95 | 11.836 | −53.05 | 46.00 | 30.77 |

The changes in the earthworms' post-test biomasses were directly proportional to the number of surviving earthworms (Figure 2), with the highest loss in Bačka soil samples with 0.3% MP (−53.05%). Furthermore, even the control group in Bačka soil (with no added MP) had a high biomass percentage loss, similar to Banat 2 with 0.3% MP (−25.93% and 26.66%, respectively), which was due to the soils' physicochemical characteristics, rather than the presence of MP particles. When the individual earthworms' post-test masses were compared, it was observed that the reduction was up to 94.827 mg per earthworm in Bačka soil with 0.3% MP. The most minimal loss was observed in the Banat 1 control group (0.094694 mg per individual), and the earthworms from the Banat 2 control group even gained weight (0.44653 mg per individual).

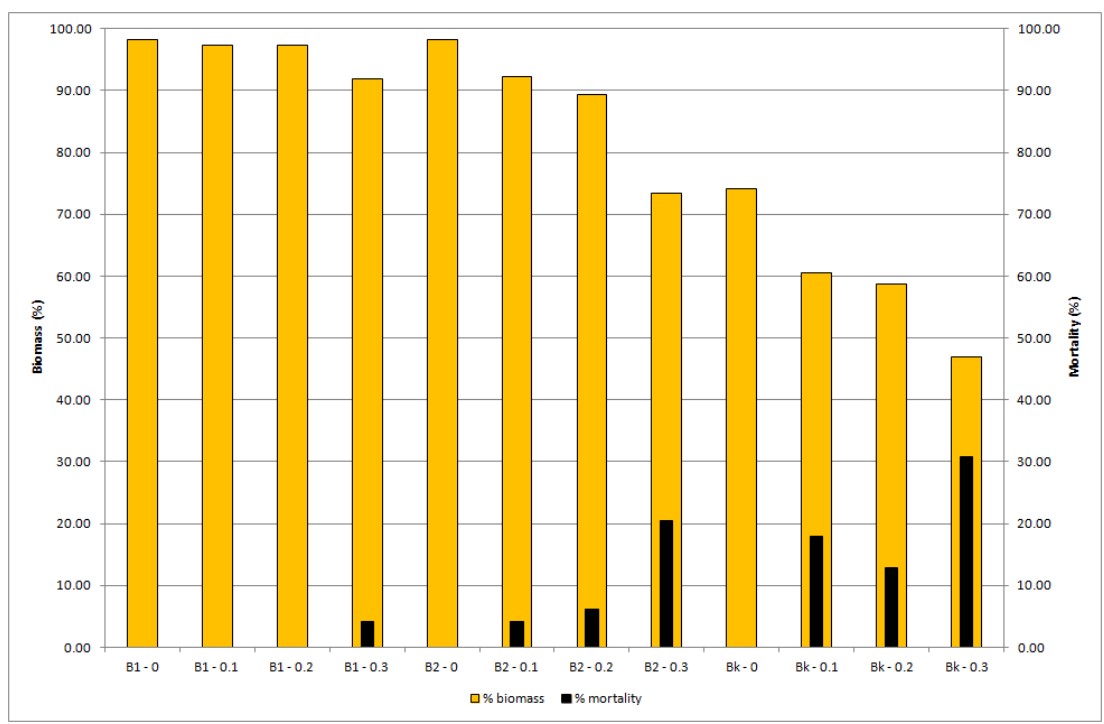

**Figure 2.** Proportional ratio of *Eisenia fetida* biomass changes (in %) and corrected mortality (%) according to Schneider-Orelli's formula (2) (B1—Banat 1; B2—Banat 2; Bk—Bačka; 0—control group; 0.1; 0.2 and 0.3—MP concentrations).

The results of one-way ANOVA regarding soil samples, MP concentrations, and replicates as the categorical predictors are presented in Table 6.

**Table 6.** The results of one-way ANOVA regarding soil samples, MP concentrations, and replicates as the categorical predictors.

| Dependent Variable | Categorical Variable | $p$ Values | Significance: * for $p < 0.05$; ** for $p < 0.01$ |
|---|---|---|---|
| Post-test number of earthworms (ptn) | Soil sample | 0.000000 | ** |
| | MP concentrations | 0.057028 | - |
| | Replicates | 0.931137 | - |
| Post-test biomass (g) (ptb) | Soil sample | 0.000000 | ** |
| | MP concentrations | 0.023188 | * |
| | Replicates | 0.700555 | - |
| Biomass change (of 100%) (bc) | Soil sample | 0.000000 | ** |
| | MP concentrations | 0.034925 | * |
| | Replicates | 0.893958 | - |
| Mortality (%) (m) | Soil sample | 0.000000 | ** |
| | MP concentrations | 0.057028 | - |
| | Replicates | 0.931137 | - |

In all four cases of dependent variables—the earthworms' post-test number, post-test biomass (in g), biomass change (of 100%), and percentage of mortality—ANOVA emphasized very high statistical differences among the three tested soil types (for $p < 0.01$). Different MP concentrations showed high statistical significances only in cases of measured and calculated biomass (Table 6). Fisher's LSD test highlighted high statistical differences regarding the highest MP concentration (0.3%) and the control groups in both cases (for $p < 0.01$, $p_{ptb} = 0.003038$, and $p_{bc} = 0.004280$, respectively); as well as 0.3% and 0.1% MP in case of the calculated percentage of biomass change (for $p < 0.05$, $p_{bc} = 0.028595$).

The results of this study are in accordance with the findings of Cao et al. [58], who proved that polystyrene microplastics (PS-MPs) obstructed the growth of *E. fetida* and, at higher exposure levels ($\geq 0.5\%$), led to a noticeable increase in mortality, suggesting that the observed effects could be attributed to the impairment of the earthworms' self-defense mechanisms. Wang et al. [59] exposed *E. fetida* to different concentrations (0, 1, 5, 10, and 20% of dry weight) of polyethylene (PE $\leq 300$ μm) particles in agricultural soil to assess oxidative stress. Using fluorescence imaging with Nile Red staining, they observed that *E. fetida* ingested PE. Furthermore, when exposed to the high concentration (20%) of PE for 14 days, there was a significant ($p < 0.05$) increase in catalase and peroxidase activity, as well as elevated levels of lipid peroxidation. Simultaneously, the activity of superoxide dismutase and glutathione S-transferase in *E. fetida* was inhibited, which clearly indicated the induced oxidative stress.

In addition to MP concentrations as a significant factor, various studies proved that the size of MP particles considerably influences earthworm avoidance behavior [14,26]. According to Jiang et al. [14], the size of 1300 nm of MPs exhibited higher toxicity and accumulated in greater quantities in earthworm intestines when compared to 100 nm particles, and they led to histopathological damage in the earthworm intestines. Chen et al. [26] concluded that MPs could have detrimental biochemical impacts on earthworms, as these soil pollutants lead to surface damage, trigger oxidative stress, and induce neurotoxic responses in *E. fetida*. The series of standardized bioassays on the biotoxicity of biodegradable (polylactic acid, polypropylene carbonate) and non-degradable PE microplastics using *E. fetida* as a bioindicator performed by Ding et al. [20] highlighted that MPs' concentration, rather than the plastic type, was more important in regulating earthworm responses to the soil contamination.

## 4. Conclusions

Since earthworms play a crucial role as edaphon members, they could be used as precise and accurate bioindicators in current agricultural soil quality assessment and future prediction. The results of this study indicate that *E. fetida* avoided the sections with MP particles. In all types of bioassays, the highest mortality rate was detected in Bačka soil samples with the highest MP concentration (0.3%), and the changes in earthworms' post-test biomasses were directly proportional to the number of surviving earthworms. Nevertheless, further studies are particularly necessary in order to obtain more detailed information on the impacts of different MP types, sizes and concentrations, as well as their combined effects with soil physicochemical characteristics and possible earthworm physiological responses, such as oxidative stress. Studying the earthworm behavioral responses to MPs and other pollutants is becoming an essential measure in evaluating the impacts of these substances on soil quality and the earthworms themselves.

**Author Contributions:** Conceptualization, M.B. and A.P.; Data curation, D.S.; Formal analysis, M.B., A.P., A.T. and T.Z.; Funding acquisition, A.P., A.T. and V.B.; Investigation, M.B.; Methodology, A.P., A.T., T.Z. and V.B.; Project administration, A.T.; Resources, M.B., A.P. and V.B.; Software, D.S.; Supervision, V.B.; Validation, T.Z., S.G. and V.B.; Visualization, M.B.; Writing—original draft, M.B.; Writing—review and editing, A.P. All authors have read and agreed to the published version of the manuscript.

**Funding:** This research received no external funding.

**Institutional Review Board Statement:** Not applicable.

**Data Availability Statement:** The datasets used and analyzed during the current study are available from the corresponding author upon reasonable request.

**Acknowledgments:** The authors would like to acknowledge the support of COST Action: Plastics monitoRIng detectiOn RemedIaTion recovery—PRIORITY, CA20101 (European Cooperation in Science and Technology).

**Conflicts of Interest:** The authors declare no conflicts of interest.

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
