# Peer review of "Effects of Polyethylene Microplastics in Agricultural Soil on Eisenia fetida (Annelida: Oligochaeta) Behavior, Biomass, and Mortality"

_agriculture, doi:10.3390/agriculture14040578_

Round 1

Reviewer 1 Report

Comments and Suggestions for Authors

The discussed manuscript is very interesting and raises the important problem of the presence of plastics in the environment. The article is written in a logical and orderly manner, but before publishing it, you should also pay attention to:

1. Abstract – perhaps these numbers of earthworms should be rounded. It looks strange if 3.8 earthworms were in the microplastic section and 6.09 in the control section. I understand that this is an average value, but it looks as if the earthworms were divided into pieces, or some earthworms were in both sections at the same time. If these are results expressed in %, this should be provided and then such values are not a problem.

2. It should be explained why these specific concentrations of microplastics were selected for testing.

3. I suggest re-reading the manuscript carefully to eliminate minor linguistic and punctuation errors.

Comments on the Quality of English Language

 I suggest re-reading the manuscript carefully to eliminate minor linguistic and punctuation errors.

Author Response

Thank you very much for taking the time to review this manuscript. Your comments and suggestions will certainly increase the quality of our manuscript. Please find the detailed responses below and the corrections in track changes in the re-submitted files.

Reviewer 2 Report

Comments and Suggestions for Authors

Overall comments

The manuscript named " Effects of polyethylene microplastics in agricultural soil on Eisenia fetida (Annelida: Oligochaeta) behavior, biomass and mortality " was reviewed. This manuscript describes the avoidance behavior, biomass changes and mortality rate of earthworm in three types of soil as affected by PE microplastics. The ecological effects of microplastics on soil health obtain great concern. The data from this study improves our understanding of the impact of microplastics on earthworm in soil. Whereas, the description and analysis of the results are not concise and explicit enough, and the readability needs to be improved. It could not be accepted for publication in its current form. Some issues should be treated.

Detailed Comments
Abstract:

1. Line 18-21: The sentence "The aim of …………………mortality rate" is hard to follow and does not properly clarify the aim of the study. Please rewrite/rephrase.

2. Line 21-22: Details of "Materials and Methods" (i.e., five replicates) should not be shown in abstract.

Introduction:

1. Line 41-43: Some details in the manuscript are incorrect, for example, the order of MP and NP is reversed. Please double check the whole manuscript.

2. Line 45-50: The authors might have tried to explain the widespread use of plastic in agriculture, but these statements don't depict it properly.

3. In the "Introduction" section, the knowledge gap is not well depicted. The authors should highlight the gaps in previous studies, which will be address in this study. Furthermore, hypotheses are needed in this section.

4. The objectives of this study should be rewrite/rephrase, which should be followed by how these objectives will be addressed in this study.

Materials and Methods:

1. What's the content of microplastics in the sampled soil in this study?

2. Line 130-132: It is unclear why the exposure dose in this study was used. Please clarify if this is an environmentally relevant dose.

3. Please add drawing software in "Statistical analyses".

4. How does the author define the significance of the data? Please add this information in "Statistical analyses".

Results and Discussion:

1. Line 205-209: How does the author define "favorable", "better"? What are the bases of these definitions?

2. Table 2, check the writing of H2O.

3. Figure 1 was not shown in the manuscript.

4. There was not sufficient justification for the exclusive focus on PE microplastic, without considering other types of plastics. In lieu of this, the author should constrain the interpretation of the results to PE microplastic. Please double check the whole manuscript.

5. Line 240-244: please specify "respond".

6. Line 246-247: How does the author define "high" in the sentence? The extra comma should be deleted. Please double check the writing in the whole manuscript.

7. Line 290-291: Is this an independent paragraph?

8. The discussion of the results lacked depth, which can be improved by developing more support for the statements and hypotheses presented via comparisons with the existing literature.

Comments on the Quality of English Language

The description and analysis of the results are not concise and explicit enough, and the readability needs to be improved. Moderate editing of English language required.

Author Response

(The authors gave the same response as above.)

Reviewer 3 Report

Comments and Suggestions for Authors

Your work is good; however, it lacks volume and nobility. Incorporating the points you recommended for future work into the current study may improve the work's quality.  Before resubmitting your paper, please consider the following issues:

·        Line 19: Why only Eisenia fetida? When there are up to 10 common earthworm species in agricultural soils that can be grouped into three ecological types: epigeic, endogeic and anecic earthworms – each group having a unique and important function.

·        Line 24: better you make it Plastic wastes

·        Line 35: make it considerably/significantly

·        Line 248: were not fed or didn’t feed?

·        Line 152: regarding or due to?

·        Line 153: change ‘present’ to ‘available’

·        It would be good if you could show us some of your experiments using a picture, as a picture is worth a thousand words.

·        Line 205: what does ‘favorable chemical properties’ mean? Favorable chemical properties for what?

·        Line 206: Why do you prefer to express soil texture as poor or good when there are more relevant phrases to use depending on soil particle distribution? 

·        Line 206: Is soil texture the only factor examined when determining if the soil is suitable for earthworms?

·        Lines 214-215: where is figure 1?

·        You stated that you replicated every experiment; however you have not shown the SD in any of your results. Why?

·        Line 334-337: It would be good and increase the volume of your work if you could do the points you recommended for the future work.

Comments on the Quality of English Language

the paper needs minor polishing 

Author Response

(The authors gave the same response as above.)

Round 2

Reviewer 2 Report

Comments and Suggestions for Authors

Please improve the English quality of the manuscript.

Comments on the Quality of English Language

Please improve the English quality of the manuscript. 

Author Response

Dear Editors and Reviewer,

Thank you very much for your letter and for the reviewers comments concerning our manuscript.

All comments were valuable and very helpful for revising and improving our manuscript. We have accepted your latest comments and have made corrections which we hope will meet with your approval. The English quality of the manuscript has been improved by the English professors employed at our Faculty. Corrected text in the article has been marked in red or green by the track changes.

Reviewer 3 Report

Comments and Suggestions for Authors

You incorporated all the comments to the level of my satisfaction.

Author Response

Dear Editors and Reviewer,

We appreciate the time and effort that the reviewer has dedicated to provide valuable feedback on our manuscript. All comments were helpful and very useful for improving our article quality and its scientific soundness.